# Metabolomic Profiles of Mouse Tissues Reveal an Interplay between Aging and Energy Metabolism

**DOI:** 10.3390/metabo12010017

**Published:** 2021-12-26

**Authors:** Qishun Zhou, Jakob Kerbl-Knapp, Fangrong Zhang, Melanie Korbelius, Katharina Barbara Kuentzel, Nemanja Vujić, Alena Akhmetshina, Gerd Hörl, Margret Paar, Ernst Steyrer, Dagmar Kratky, Tobias Madl

**Affiliations:** 1Gottfried Schatz Research Center for Cell Signaling, Metabolism and Ageing, Molecular Biology and Biochemistry, Medical University of Graz, 8010 Graz, Austria; qishun.zhou@medunigraz.at (Q.Z.); jakob.kerbl-knapp@medunigraz.at (J.K.-K.); fangrongzhang@fjmu.edu.cn (F.Z.); m.korbelius@medunigraz.at (M.K.); katharina.kuentzel@medunigraz.at (K.B.K.); nemanja.vujic@medunigraz.at (N.V.); alena.akhmetshina@medunigraz.at (A.A.); ernst.steyrer@medunigraz.at (E.S.); dagmar.kratky@medunigraz.at (D.K.); 2Key Laboratory of Gastrointestinal Cancer (Fujian Medical University), Ministry of Education, Fuzhou 350001, China; 3Otto-Loewi Research Center, Physiological Chemistry, Medical University of Graz, 8010 Graz, Austria; gerd.hoerl@medunigraz.at (G.H.); margret.paar@medunigraz.at (M.P.); 4BioTechMed-Graz, 8010 Graz, Austria

**Keywords:** aging, NMR spectroscopy, mice, energy metabolism, fat, intestine, metabolomics

## Abstract

Energy metabolism, including alterations in energy intake and expenditure, is closely related to aging and longevity. Metabolomics studies have recently unraveled changes in metabolite composition in plasma and tissues during aging and have provided critical information to elucidate the molecular basis of the aging process. However, the metabolic changes in tissues responsible for food intake and lipid storage have remained unexplored. In this study, we aimed to investigate aging-related metabolic alterations in these tissues. To fill this gap, we employed NMR-based metabolomics in several tissues, including different parts of the intestine (duodenum, jejunum, ileum) and brown/white adipose tissues (BAT, WAT), of young (9–10 weeks) and old (96–104 weeks) wild-type (mixed genetic background of 129/J and C57BL/6) mice. We, further, included plasma and skeletal muscle of the same mice to verify previous results. Strikingly, we found that duodenum, jejunum, ileum, and WAT do not metabolically age. In contrast, plasma, skeletal muscle, and BAT show a strong metabolic aging phenotype. Overall, we provide first insights into the metabolic changes of tissues essential for nutrient uptake and lipid storage and have identified biomarkers for metabolites that could be further explored, to study the molecular mechanisms of aging.

## 1. Introduction

Aging affects most living organisms and can be manifested as time-dependent accumulation of structural and functional alterations in an organism [1]. During aging, the accumulation of cellular damage leads to the disruption of regular physiology and functional impairments, and can promote cellular senescence [2]. This process can be understood as a series of simultaneous biological activities, occurring in the body at the cellular, tissue, organ, or systemic level. Aging is characterized by nine hallmarks: genomic instability, telomere attrition, epigenetic alterations, loss of proteostasis, deregulated nutrient-sensing, mitochondrial dysfunction, cellular senescence, stem cell exhaustion, and altered intercellular communication [1], all of which are associated with adverse effects on metabolism and biological functions [1,3]. Aging is associated with an increasing incidence of numerous diseases and pathologies, such as cardiovascular diseases, including atherosclerosis; cancer; geriatric syndromes; type 2 diabetes; and the co-occurrence of chronic diseases [4,5,6,7,8]. Despite the fact that the exact biological mechanisms of aging are poorly understood, studies have been performed to identify metabolic biomarkers of aging derived from either human data or mouse models [9,10,11,12]. In these studies, modern metabolomics techniques, such as mass spectrometry and nuclear magnetic resonance (NMR) spectroscopy have been used to efficiently characterize the metabolome of an organism [9], reflecting alterations in small molecule content following genomic, transcriptomic, and proteomic changes. To date, more than 100,000 identified or predicted human metabolites have been documented [13]. Given that different tissues serve distinct metabolic functions and may age differently [12], it is essential to compare various tissues of the same organism at different ages to reveal the concomitant variations in metabolism at different levels (cell, tissue, organ) within an organism.

Among all the complex biological mechanisms in humans, energy metabolism is closely related to the aging process; as the basal metabolic rate decreases linearly with age [14]. Multiple studies have identified the potential effects of aging on energy intake and expenditure [14,15,16]. It has been suggested that essential metabolites of energy metabolism, including ATP (AMP), malondialdehyde, succinate, or pyruvate, play an important role or are affected during aging [16,17,18]. Life span is considered to be closely linked to energy metabolism [19]. However, systematic studies investigating the changes in metabolites in different energy metabolism-related organs and tissues during aging and how the body controls them as a whole are still scarce [20,21]. Therefore, we aimed to perform such a systematic study, addressing the metabolome of tissues involved in energy metabolism to identify potential biomarkers of aging. Based on the analogy between humans and mice, we selected plasma, skeletal muscle, brown adipose tissue (BAT), white adipose tissue (WAT), and small intestine (duodenum, jejunum, ileum) from young and old mice to predict human biomarkers of aging.

Different AT depots have distinct functions, with the WAT representing the main energy storage depot, whereas the BAT dissipates energy in the form of heat [22]. The aging process is accompanied by a decrease in BAT mass and activity, as well as a reduced browning of the WAT [23,24]. The amount of WAT initially increases with age, but is redistributed in different locations during aging [25]. Changes in BAT and WAT function have been associated with metabolic abnormalities, including insulin resistance, chronic inflammation, or increased risks of diabetes [26]. Therefore, understanding the metabolic alterations behind the structural and functional changes in adipose tissue depots will shed light on the underlying mechanisms. Aging has also been associated with reduced nutrient absorption and altered morphology of the small intestine [27]. As for skeletal muscle, aging causes a loss of mass and function over time [28], which has been connected to several phenomena, including changes in protein synthesis and degradation, a decreased number of motor axons, increased reactive oxygen species (ROS), and mitochondrial dysfunction [28]. Plasma, as an essential extracellular fluid in the body that transports proteins, nutrients, hormones, and metabolic waste destined for excretion, plays an irreplaceable role in energy metabolism [29]. In addition, plasma biomarkers, either in the form of proteins [30] or metabolites [31], have been linked to aging. 

Overall, the above-mentioned organs and tissues have been shown to be functionally altered during aging. We, therefore, used untargeted NMR metabolomics in the aforementioned tissues to identify novel metabolite biomarkers of aging on a tissue-specific and systemic level. Furthermore, we highlighted their potential implications on energy metabolism. NMR-based metabolomics provided an adequate approach to experimentally characterize our samples with a high quantification precision, while keeping the samples intact. In addition, it enabled an automated workflow and a relatively short process time for a large number of samples [32]. Using metabolic profiling of essential organs from young and aged mice, we have already identified multiple metabolites, including several amino acids (AAs), 4-aminobutyrate, succinate, nicotinamide, and other small molecules as tissue specific biomarkers of aging [20]. The present study focuses on tissues associated with energy metabolism in young and aged mice, with the aim of gaining a comprehensive insight into metabolic profiles from both tissue-specific and systemic perspectives. Our results indicated the following tissue- and plasma-specific biomarkers of aging: lactic acid, citric acid, and glucose in plasma; arginine, glycine, inosinic acid, and phenylalanine in skeletal muscle; phosphorylcholine, lysine, and hypoxanthine in BAT; citric acid in WAT; allantoin and inosine in ileum. Citric acid and lysine are biomarkers of aging across tissues. Combined with our previous results in brain, heart, kidney, liver, lung, and spleen [20], we identified tyrosine as a global biomarker of aging, as it was systematically changed in seven different tissues. This study provides a general overview of the altered metabolic profiles of various metabolically active tissues in aging and highlights potential biomarkers associated with this process. It also provides robust evidence of the biological mechanisms of aging from an additional dimension, which could be used either to identify targets for anti-aging pharmacology or as indicators to validate the effect of senolytic therapies.

## 2. Results

Our goal was to identify the tissue-specific metabolomic profiles of young and aged mice, to shed light on their respective metabolic alterations during aging. Moreover, we aimed to elucidate the interrelationships and dependencies between tissue metabolism and the systemic metabolic signatures of aging. To systematically characterize the aging process of each tissue, we used an untargeted NMR-based strategy, as previously reported [20], and performed metabolic profiling of plasma, skeletal muscle, BAT, WAT, duodenum, jejunum, and ileum from young and old mice. 

### 2.1. Aging-Associated Changes in Plasma Are Closely Related to Energy Metabolism

Plasma is composed of several components that change as aging progresses. Interestingly, it has been found that replacing half of the plasma of mice with saline containing 5% albumin can improve aging-related changes in mice [33], suggesting that compounds in plasma may contribute to aging. Plasma samples from young and aged mice showed a predictive score (T score [1]) of 35.1% on the O-PLS-DA plot, indicating that they are well separated (Figure 1A, left, middle). Resampling analysis through permutation test, which exchanged labels on the data points resulted in R^2^Y at 0.99 (*p* = 0.01) and Q^2^ at 0.902 (*p* = 0.01) (Figure 1A, right). This clearly indicates that the difference is unlikely to originate from a random distribution, i.e., this highlights its significance. Moreover, analysis of the metabolites in plasma revealed increases in acetone, citric acid, lysine, creatine, myo-inositol, threonine, and indoxyl sulfate, and decreases in valine, lactic acid, alanine, acetic acid, glycerol, and glucose during aging (Figure 1B and Appendix A). Among these metabolites, the levels of citric acid, indoxyl sulfate, threonine, myo-inositol and acetone increased by more than two-fold, while, those of lactic acid and acetic acid decreased more than two-fold. Notably, indoxyl sulfate showed the highest increase (5.2-fold), and acetic acid decreased the most (2.1-fold). To further support our findings, we performed metabolite enrichment analysis (MSEA) [34] and matched the significantly altered metabolites to a metabolic pathway library [35], to identify metabolic pathways associated with the observed age-related changes in mice. These alterations in plasma were mainly related to galactose metabolism, glucose-alanine cycle, transfer of acetyl groups to mitochondria, Warburg effect, and glycine and serine metabolism (Figure 1C). 

### 2.2. Metabolic Changes in Skeletal Muscle during Aging

It has been previously shown that aging exacerbates skeletal muscle atrophy [36] and that biomarkers of aging are abundantly expressed in this tissue [21,37]. The NMR profiles of skeletal muscle samples from young and old mice were clearly distinguishable from each other in the O-PLS-DA plot, with a T score [1] of up to 44.4%; overwhelmingly higher than the one of the orthogonal component (Figure 2A, left, middle). A permutation test yielding R^2^Y at 0.954 (*p* = 0.03) and Q^2^ at 0.712 (*p* = 0.02) indicated the significant separation of the two groups (Figure 2A). Accordingly, we identified a large number of metabolites that were altered in the muscles of aged mice, including a higher abundance of leucine, isoleucine, arginine, glycogen, inosine, tyrosine, and phenylalanine, and a lower abundance of lactic acid, alanine, acetic acid, creatine, glycine, glucose, inosinic acid, and fumaric acid (Figure 2B and Appendix A). The levels of these metabolites were increased/decreased by less than two-fold. These significantly altered metabolites were mainly related to aspartate metabolism, phenylalanine and tyrosine metabolism, arginine and proline metabolism, urea cycle, and glycine and serine metabolism (Figure 2C).

### 2.3. BAT and WAT Age Differently

BAT displays a decreased activity, and redistributes in the human body, during aging [24]. In addition, the concomitant metabolic dysfunction is caused by alterations in WAT and/or strongly affects WAT homeostasis [38]. Similar to other tissues, the metabolic fingerprints of BAT from young and old mice demonstrated a distinct heterochronic clustering of the tissues, with the T score of the predictive component at 29.4% (Figure 3A, left, middle). The permutation test showed a correlation coefficient of R^2^Y at 0.992 (*p* < 0.01) and Q^2^ at 0.82 (*p* < 0.01) for BAT, indicating the significance of the difference (Figure 3A, right). In contrast, we found for the O-PLS-DA of WAT a T score [1] at 19.4%. This indicated an insignificant separation of the two groups (Figure 4A, left, middle); as well as the permutation test showing a R^2^Y of 0.999 (*p* < 0.01) and Q^2^ of 0.624 (*p* = 0.16) for WAT, further indicating the insignificance of alteration (Figure 4A, right). In the volcano plot and the reduced spectra of BAT, we found more isoleucine, leucine, valine, methionine, citric acid, lysine, threonine, tyrosine, histidine, tryptophan, and phenylalanine, as well as less alanine, phosphorylcholine, and hypoxanthine in BAT during aging (Figure 3B and Appendix A). Changes in metabolite levels were below two-fold. The significantly altered metabolites in BAT were mainly related to glycine and serine metabolism; valine, leucine, and isoleucine degradation; phenylalanine and tyrosine homeostasis; and methylhistidine, and biotin metabolism (Figure 3C). However, the metabolites in WAT were less affected by aging, since only the higher abundance of citric acid and lower abundance of myo-inositol were significant. The levels of citric acid increased 2.1-fold in WAT and 2.0-fold in the BAT of aged mice, respectively (Figure 4B and Appendix A).

### 2.4. Small Intestine Is Metabolically Inert during Aging

To complete our systemic screening on the consequences of aging in metabolically active tissues, we determined the metabolic signatures of duodenum, jejunum, and ileum of young and old mice. It has been reported that the number of endocrine cells in the duodenum changes during aging [39], whereas data on time-dependent alterations in the jejunum are scarce [27]. However, morphological modifications and impaired function were observed in the ileum or at the ileocecal junction at an advanced age [27,40]. We obtained a T score [1] at 11.9% (Figure 5A, left, middle), a correlation coefficient R^2^Y at 0.986 (*p* = 0.08) and Q^2^ at 0.374 (*p* = 0.30) for duodenum (Figure 5A, right), and NMR-based metabolomics profiling of duodenal metabolites only revealed a decrease in lysine during aging (Figure 5B and Appendix A), despite a trend toward a reduction in all metabolites. Jejunal samples showed a T score [1] at 16.2% (Figure 5A, left, middle), with a permutation correlation coefficient R^2^Y of 0.935 (*p* = 0.29) and Q^2^ of 0.413 (*p* = 0.16) (Figure 6A, right). The volcano plot and reduced spectra indicated no significantly altered metabolites in the aging jejunum (Figure 6B and Appendix A). The ileum showed significant alterations during aging, despite a T score [1] at 12.2% (Figure 7A, left, middle), which was associated with a permutation correlation coefficient R^2^Y at 0.997 (*p* < 0.01) and Q^2^ of 0.855 (*p* = 0.01) between the clusters of samples from young and old mice (Figure 7A, right). In the volcano plot and reduced spectra, an increase of dimethylamine and allantoin and a decrease of inosine were evident, among which the changes of allantoin and inosine levels were 2.8-fold (increase) and 2.3-fold (decrease), respectively (Figure 7B and Appendix A).

### 2.5. Overview

We summarized the variation of all metabolites from the seven studied tissues in a heat map, shown in Figure 8. This highlights the metabolic changes of tissues, and additionally reveals the enrichment of amino acids in the intestines, and the increase of metabolites related to the tricarboxylic acid cycle (TCA cycle), glycolysis/gluconeogenesis, and amino acid derivatives in the plasma, skeletal muscle, BAT, and WAT.

## 3. Discussion

The accumulation of abnormalities and loss of functional mass in organs as a result of aging are closely related to altered energy metabolism and expenditure [5,41,42]. The study of tissues responsible for energy uptake, storage, and utilization can provide essential information about the corresponding metabolic dysregulations. The analysis of murine small intestine, adipose tissue, muscle, and plasma by untargeted NMR-based metabolomics allowed us to identify the respective metabolite composition with high confidence. This shed light on the disturbances in energy homeostasis, which also explain the observed metabolic changes in aging tissues. 

The plasma metabolome has been extensively studied in recent years, to identify biomarkers or to predict biological age, the rate of aging, and age-related disease risks [31,43,44,45,46]. In our study, we observed a consistent increase in plasma myo-inositol and a decrease in glycerol. In humans, conflicting results have been reported, with either increased [43] or decreased [31] aging-associated plasma citric acid levels. A reduction in plasma citric acid levels has been proposed to be a consequence of diminished citrate synthase activity with aging, resulting in higher citric acid concentrations in younger populations [31]. However, and in accordance with other human data [43], we found that citric acid was strongly increased in the plasma of aged mice. The increase in citric acid in aged humans and mice may be a consequence of decreased aconitase activity, which has been reported for several tissues [47,48], and which would lead to reduced citric acid degradation. Both mechanisms affect the citric acid cycle and cause variations in citric acid concentrations in aged humans and mice [47], yielding apparently-contradictory results. A different set of AAs were found to be significantly altered in the plasma of elder humans and aged mice, with the exception of serine, which decreases in both [43,46]. Our studies revealed a consistent result with previous mice experiments, with respect to a decrease in alanine [46]. In addition, we identified a decrease of valine and an increase in lysine and threonine, indicating both similarities and differences between studies. Potential reasons for differences could be age of mice (9–10 weeks vs. 24 weeks), or different mouse strains (mixed genetic background of 129/J and C57BL/6 vs. C57BL/6J). Thus, follow-up studies to address the potential impact of these variables on the plasma metabolome of mice will be necessary. The BCAAs valine, leucine, and isoleucine in plasma have been suggested as indicators of fasting, because they are used as energy sources [44]. Thus, their decreased concentrations in aged mice may suggest a reduced mobilization of endogenous energy sources. Notably, we also found decreased glucose concentrations in aged mice, which contrasts with its increase found in aging humans in previous studies [43,49], which may be due to the high-calorie diet in developed regions. In addition, we observed an increased level of myo-inositol in plasma. Myo-inositol is a precursor of phosphatidylinositol-3-phosphate (PI3P) [50,51], which activates autophagy. Autophagy is impaired during aging [52], and myo-inositol, per se, is a promoter of longevity [53]. Hence, the elevated myo-inositol concentrations in our mice might be the result of reduced conversion into PI3P, indicating a reduced activity of several enzymes targeting myo-inositol or its downstream products. Overall, our results revealed a set of aging biomarkers in plasma consistent with some previous studies [43,46], but also a few contrasts to other studies [31,46,49]; implying the heterogeneity of plasma in different species and populations.

The metabolome of skeletal muscles has also been studied in detail [37,46,54,55]. Our results showed consistent changes in the levels of leucine, lactic acid, alanine, glycine, inosine, and phenylalanine. Among the significantly altered metabolites, the increase in leucine and isoleucine may correspond to the decrease in BCAA catabolism in muscle atrophy [56]. Since it has been suggested that BCAAs stimulate muscle protein synthesis, a decline in muscle function despite their accumulation suggests a defect in the downstream processes of BCAA-stimulated protein expression [57]. The decrease in lactic acid may be a result of reduced physical activity [58], which is consistent with the decrease in alanine, a byproduct of AA catabolism during muscle catabolism [59]. Arginine has a protective effect on muscle protein synthesis [60]. Thus, more arginine in old mice may indicate a self-protective mechanism against muscle atrophy. Decreased acetic acid, creatine, and glycine levels in old mice were consistent with a reduced capacity of the muscle to generate energy [61,62,63]. The levels of lactic acid and glucose were found to increase in the skeletal muscle of aged mice [46], which is in contrast to our study. This might be due to the aforementioned differences between the mouse studies, but also indicates that these variables need to be evaluated in the future. Increased glycogen concentrations are possibly due to a decrease in glycogen hydrolysis, due to the decrease of glycogen phosphorylase activity [64]. This is in line with a previous study reporting decreased glycogen phosphorylase activity in the tibialis anterior (TA) muscle of aged rats [65]. The increased inosine level, along with decreased inosinic acid level, suggested an enhanced degradation of inosinic acid, if not all adenosine nucleotides [66,67]. Overall, our biomarkers revealed a consistent picture of the decline in muscle activity with age.

Adipose tissues of human and mice have various heterogeneities [68,69], here we aimed to decipher the possible common mechanisms of aging despite these diversities. First, we have found that most biomarkers for aging are found in BAT, including eight essential AAs and two nonessential AAs. Other biomarkers are citric acid, phosphorylcholine, and hypoxanthine. A significant amount of BCAAs is degraded in adipose tissue [70,71]. Increased BCAAs and the reduction of alanine in aged BAT, similarly to skeletal muscle, may indicate a decrease in BCAA degradation and may be related to aging-related diminished BAT thermogenesis [56,72]. This might be due to the downregulation of the BCAA aminotransferase during aging [73,74]. The methionine increase, however, might be linked to the increase in lipogenesis [75], while the accumulation of citric acid is consistent with the reduced activity of aconitase during aging [47], as observed in plasma. Interestingly, threonine and valine have been found to promote aging via activation of the TOR/S6K pathway [76]. Threonine could additionally alter metabolic homeostasis via the liver-derived hormone fibroblast growth factor 21 [77], which probably explains the increase in threonine and valine concentrations in the aging BAT. The elevated level of tyrosine might be related to the decreased synthesis of catecholamines [78].

Although not determined in this study, it has been shown that aging-induced body mass gain in female mice is associated with increased WAT but not BAT mass [25]. The authors also showed that female mice exhibit a progressive age-dependent loss of subcutaneous and visceral WAT browning, whereas BAT changes toward a fat storage phenotype. Despite these differences in WAT browning, function, and depot capacity during aging in humans and mice [24,25,38], the metabolome of WAT was comparable between young and old mice. The mostly unaffected metabolome of WAT indicates that aging has little effect on its metabolism, despite the fact that alterations in WAT function or depot capacity have been observed in humans and mice [24,38]. Less myo-inositol in WAT suggests a lack of autophagy, which may consequently promote inflammation [79]. A higher amount of citric acid in BAT, WAT, and plasma, including other tissues studied previously [20], indicates that citric acid could be a suitable biomarker for changes in energy and lipid metabolism during aging. Future studies should focus on the activities of aconitase and citrate synthase [80], in order to complement the metabolomics results. In line with this, previous studies outlined the critical role of the TCA cycle in aging [81], leading to a potential reduction in functional (including endocrine) activities [82]. Overall, we observed a concomitant and similar change in lactic acid, alanine, acetic acid, citric acid, lysine, threonine, and glucose concentrations in plasma, skeletal muscle, and BAT. This may indicate that similar mechanisms of aging occur in different tissues, which is largely consistent with our previous results [20].

According to our results, the duodenum, jejunum, and ileum remained unaffected by aging, from a metabolomic point of view. It has been suggested that aging promotes inflammation and alterations in cell function in the intestine [83], as well as malabsorption, or the reverse, increased absorption, of multiple nutrients [27]. Our results, however, indicate that the metabolic alterations corresponding to such changes may not necessarily take place. This could be due to epithelial homeostasis including rapid self-renewal [84]. Further studies are needed to validate whether the human gut has the same ‘unaged’ metabolome and to clarify whether it is common for aged mice to have unaffected intestinal nutrient absorption. In the ileum, we observed less inosine and more allantoin. Since allantoin is the metabolic product of inosine in mice [85,86], this suggests increased catabolism of adenosine and probably other nucleotides. Elevated dimethylamine levels, however, are a consequence of increased activity of intestinal bacteria that convert choline to dimethylamine [87,88]. Overall, our data imply that the duodenum, jejunum, and ileum undergo few functional changes over the entire life span. 

In addition to current knowledge on the altered metabolome in aged plasma and skeletal muscle, we have identified heterogeneity in circulating AAs in plasma of aged mice, either among individuals or between species. Elevated citric acid levels in plasma and adipose tissues confirmed the critical role of the TCA cycle in linking energy metabolism, lipid metabolism, and aging [89]. Metabolic profiling of skeletal muscle and BAT provided an overall picture of their aberrant activity. In contrast, the metabolomes of WAT and intestine remain relatively unaltered during aging. The consistency of several results from plasma and skeletal muscle with previous studies demonstrates the reliability and precision of this NMR-based metabolomics approach. Nevertheless, future studies are needed to reveal potential metabolic differences between mouse strains and metabolic changes occurring during the early life of mice in the future.

## 4. Materials and Methods

### 4.1. Collection of Mouse Tissues

Animals, diet, and mouse tissue collection have been described previously [20]. All experiments were performed in accordance with the European Directive 2010/63/EU and approved by the Austrian Federal Ministry of Education, Science, and Research.

### 4.2. NMR Sample Preparation

Thirty to fifty mg of each organ and tissue sample from wild type female mice with mixed genetic background of 129/J and C57BL/6 were resected and snap-frozen in liquid nitrogen, followed by storage at −80 °C until analysis. For metabolite extraction, 400 μL ice-cold methanol and 200 μL MilliQ H_2_O were added to each sample (except for plasma, where only 400 µL ice-cold methanol was added to 200 µL plasma). Tissue samples were transferred to 2 mL tubes with O-ring caps containing Precellys beads (1.4 mm zirconium oxide beads, Bertin Technologies, Villeurbanne, France) for homogenization by Precellys 24 tissue homogenizer (Bertin Technologies, Montigny-le-Bretonneux, France). After centrifugation at 13,000 rpm for 45 min (4 °C), the supernatant was transferred to new 1.5 mL tubes and subsequently lyophilized at <1 Torr, 850 rpm, 25 °C for 10 h in a vacuum-drying chamber (Savant Speedvac SPD210 vacuum concentrator) with an attached cooling trap (Savant RVT450 refrigerated vapor trap) and vacuum pump (VLP120) (all Thermo Scientific, Waltham, MA, USA). For the NMR experiments, the samples were re-dissolved in 500 μL of NMR buffer (0.08 M Na_2_HPO_4_, 5 mM TSP (3-(trimethylsilyl) propionic acid-2,2,3,3-d4 sodium salt), 0.04 (*w*/*v*)% NaN_3_ in D_2_O, pH adjusted to 7.4 with 8 M HCl and 5 M NaOH). For BAT, WAT, and intestinal samples, 50 µL of chloroform (CHCl_3_) was added to the solution in NMR buffer then centrifuged at 13,000 rpm for 10 min (4 °C) to remove lipids. The supernatants were transferred to 5 mm NMR tubes for data acquisition.

### 4.3. Data Acquisition and Analysis

Metabolic-profiling analysis was conducted at 310 K using a 600 MHz Bruker Avance Neo NMR spectrometer (Bruker Biospin, Rheinstetten, Germany) equipped with a TXI 600S3 probe head. The Carr–Purcell–Meiboom–Gill (CPMG) pulse sequence was used to acquire ^1^H 1D NMR spectra with a pre-saturation for water suppression (cpmgpr1d, 512 scans, 73,728 points in F1, 12019.230 Hz spectral width, 1024 transients, recycle delay 4 s) [90,91]. The ^1^H-^13^C heteronuclear single-quantum correlation (HSQC) spectra were recorded with a recycle delay of 1.0 s, spectral widths of 20.8/83.9 ppm, centered at 3.9/50.0 ppm in ^1^H/^13^C, with 2048 and 256 points, respectively, and 8 scans per increment. NMR spectral data were processed as previously described [92]. Briefly, data were processed in Bruker Topspin version 4.0.2 using one-dimensional exponential window multiplication of the FID, Fourier transformation, and phase correction. The NMR data were then imported into Matlab2014b; TSP was used as the internal standard for chemical-shift referencing (set to 0 ppm); regions around the water, TSP, and methanol signals (also CHCl_3_ signals for BAT, WAT and intestines, as well as EDTA for plasma) were excluded. The NMR spectra of two WAT samples of old mice could not be analyzed due to problems with field homogeneity. The NMR spectra were aligned, and a probabilistic quotient normalization was performed. Reduced spectra and normalized spectra were generated by Matlab2014b, among which the normalized spectra were used to quantify metabolites by signal integration. For each quantified metabolite, a characteristic peak without interfering signals was selected, the start and end points limiting the range of the peak were defined to integrate the area of the peak by summing the values for each point. To visualize our integration approach, the characteristic peaks of selected metabolites are shown in Appendix A, with the areas of integration defined by the black bars. The integrations were used to generate the orthogonal partial least squares discriminant analysis (O-PLS-DA), permutation analysis, volcano plot, MSEA (including associated data consistency checks and cross-validation), and the heat map using MetaboAnalyst 5.0 [35]. The statistical significance of the identified differences was validated by the quality assessment statistic Q^2^. A univariate statistical analysis was carried out with aforementioned integrations using GraphPad Prism 7.04 (GraphPad Software, La Jolla, CA, USA). Data were represented as mean ± standard deviation (SD). The *p*-values were calculated using a two-tailed Student’s t-test for pairwise comparison of variables. Metabolites with *p* < 0.05 are shown in panel B of each Appendix A. Additionally, we recorded for each tissue an 2D ^1^H-^13^C HSQC spectra, to highlight the presented metabolites and to validate our interpretations on 1D ^1^H spectra. These figures are included in the Appendix A.

## 5. Conclusions

Biomarkers provide essential information about the health state or disease development in patients [93]. Therefore, it is of utmost interest to identify new biomarkers in a reproducible manner and to reveal their underlying metabolic pathways. Metabolomics is a recently developed technique that allows the precise identification of molecules from biological samples [94]. In the present study, we used NMR-based metabolomics to give an overview of tissues associated with energy metabolism and highlighted several molecules related to major pathways of energy homeostasis. In addition, we identified the WAT, duodenum, jejunum, and ileum as mainly ‘unaltered’ organs, which are likely to present a lower spectrum of functional changes during aging. This leads to the hypothesis that aging mainly affects energy expenditure by reducing it, with less effects on the absorption and storage of energy. This is the first systematic study on the metabolic landscape of organs involved in energy metabolism, with the potential, in future studies, for new strategies to treat diseases related to aging and nutrition. 

## Figures and Tables

**Figure 1 metabolites-12-00017-f001:**
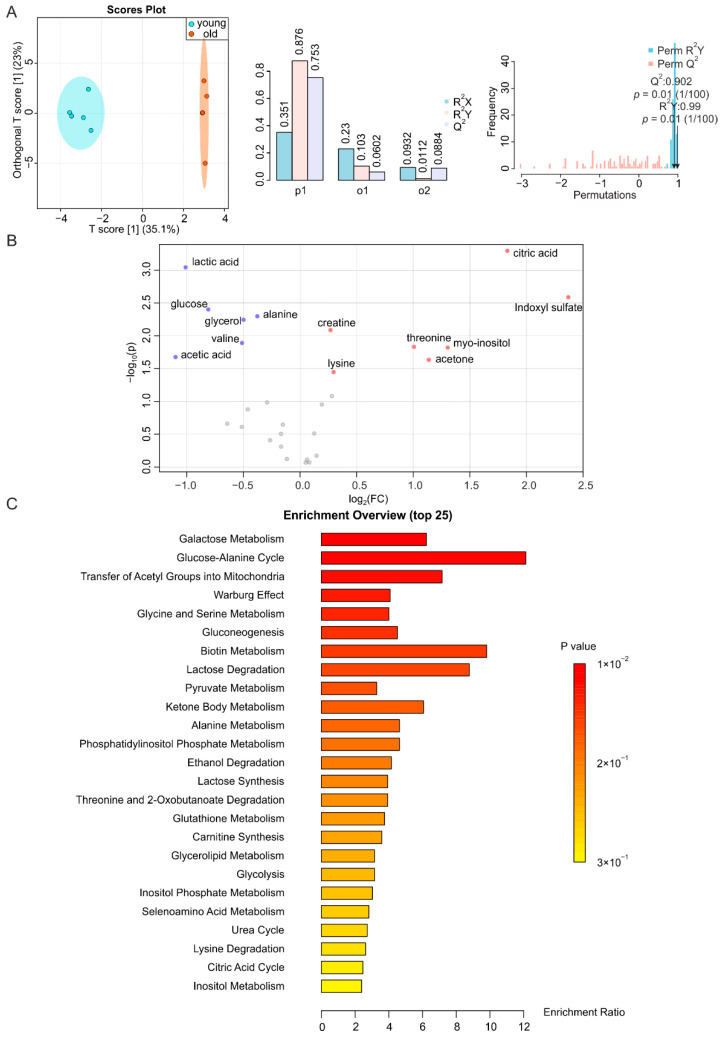
NMR-based metabolomics analysis of plasma. (**A**) O-PLS-DA plot of plasma samples from young (blue) and old (orange) mice (left), with *n* = 5 each, and the permutation between predictive component (p1) and two orthogonal components (o1, o2) with cross-validated R^2^X, R^2^Y, and Q^2^ coefficients (middle), and histogram showing the permutation test with permutation number *n* = 100 (right). Note in the O-PLS-DA plot (left panel), the 2 data points with an orthogonal T-score of 0 overlap. (**B**) Volcano plot of metabolites with different abundances between plasma from young and aged mice. Increased (red) and decreased (blue) metabolites illustrate significant fold changes during aging, while grey dots denote insignificantly changed metabolites. (**C**) Metabolite set enrichment analysis (MSEA) of affected metabolites pointing out their physiological relevance.

**Figure 2 metabolites-12-00017-f002:**
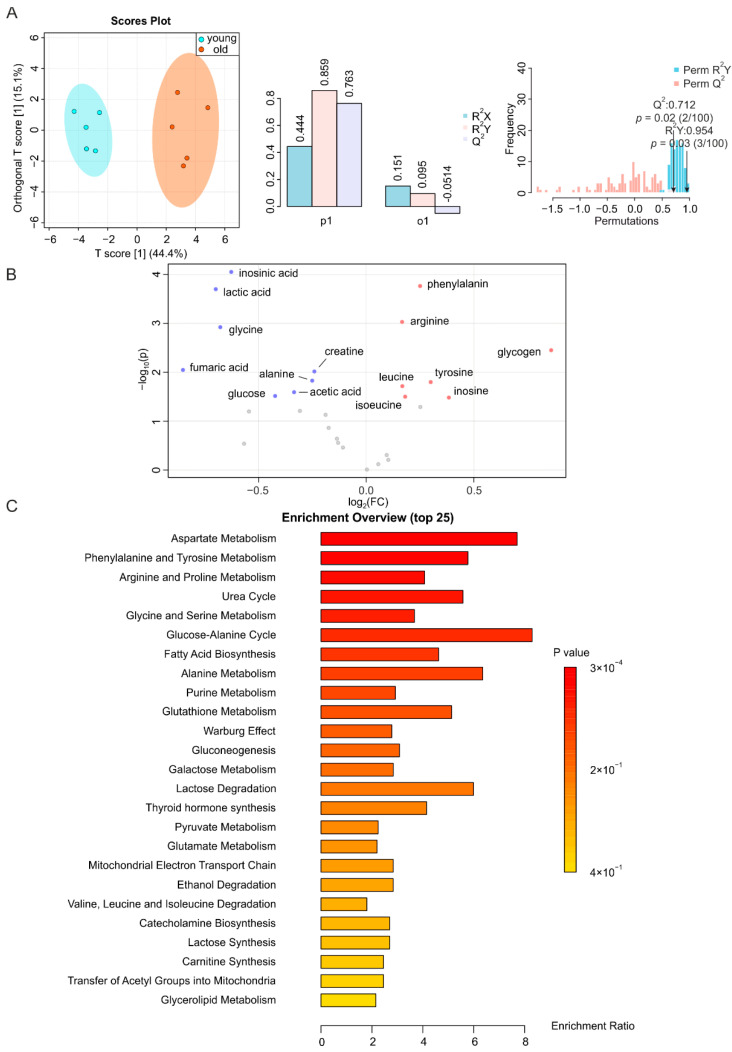
NMR-based metabolomics analysis of skeletal muscle. (**A**) O-PLS-DA plot and cross validation of skeletal muscle samples from young (blue) and old (orange) mice (left), with *n* = 5 each, and permutation between the predictive component (p1) and orthogonal components (o1) with cross-validated R^2^X, R^2^Y, and Q^2^ coefficients (middle), and histogram showing the permutation test with permutation number *n* = 100 (right). (**B**) Volcano plot of differentially regulated metabolites in skeletal muscle from aged and young mice. Increased (red) and decreased (blue) metabolites delimitate the fold change over aging, while grey dots denote insignificantly changed metabolites. (**C**) MSEA of affected metabolites in search of their functional relevance.

**Figure 3 metabolites-12-00017-f003:**
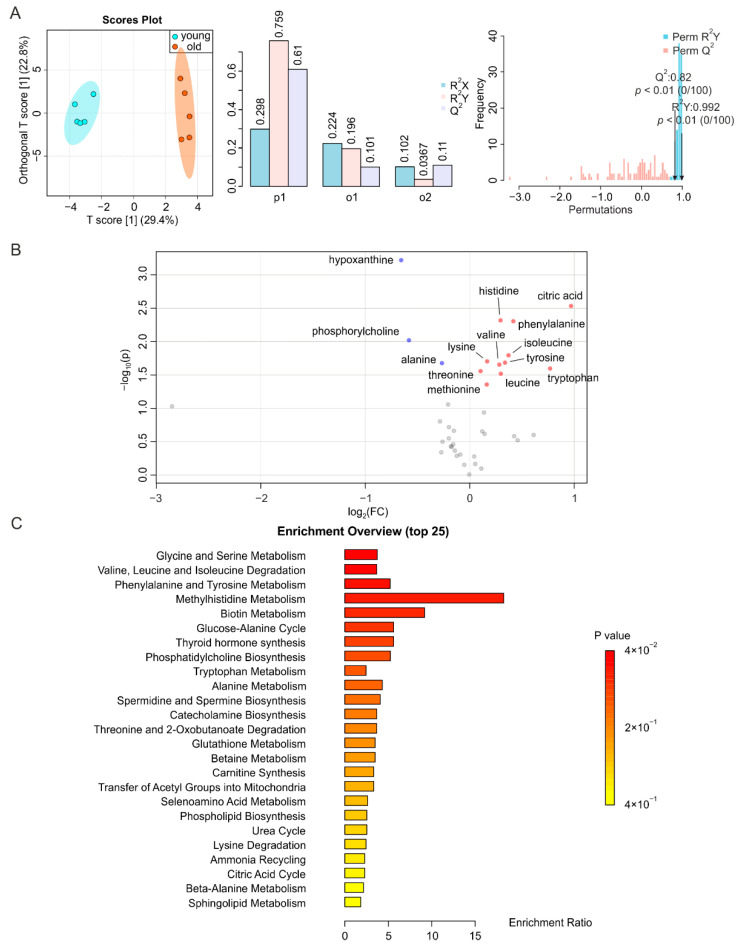
NMR-based metabolomics analysis of BAT. (**A**) O-PLS-DA plot and cross validation of BAT samples from young (blue) and old (orange) mice (left), with *n* = 5 each, and permutation between predictive component (p1) and two orthogonal components (o1, o2) with cross-validated R^2^X, R^2^Y and Q^2^ coefficients (middle), and histogram showing the permutation test with permutation number *n* = 100 (right). (**B**) Volcano plot of differentially regulated metabolites in BAT from young and aged mice. Increased (red) and decreased (blue) metabolites delimitate the fold change over aging, while grey dots denote insignificantly changed metabolites. (**C**) MSEA of affected metabolites, in search of their functional relevance.

**Figure 4 metabolites-12-00017-f004:**
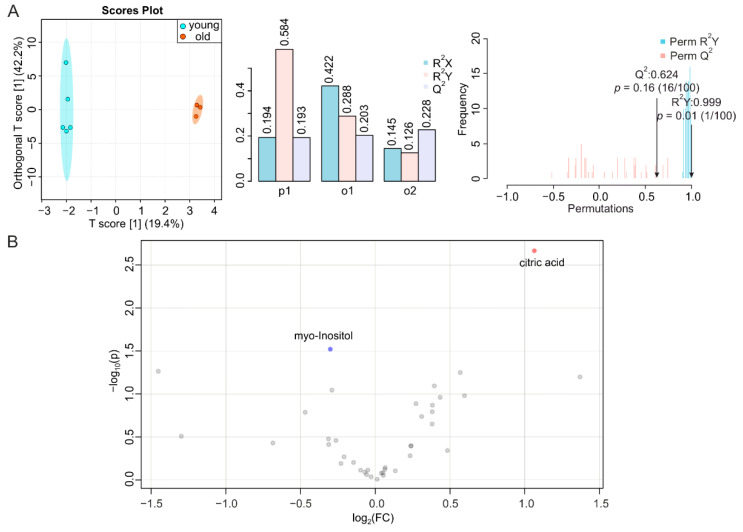
NMR-based metabolomics analysis of WAT. (**A**) O-PLS-DA plot and cross validation of WAT samples from young (blue) and old (orange) mice (left), with *n* = 5 for young and *n* = 3 for old, and permutation between predictive component (p1) and two orthogonal components (o1, o2) with cross-validated R^2^X, R^2^Y, and Q^2^ coefficients (middle), and histogram showing permutation test with permutation number *n* = 100 (right). (**B**) Volcano plot of differentially regulated metabolites in WAT from young and aged mice. Increased (red) and decreased (blue) metabolites delimitate the fold change with aging, while grey dots denote insignificantly changed metabolites.

**Figure 5 metabolites-12-00017-f005:**
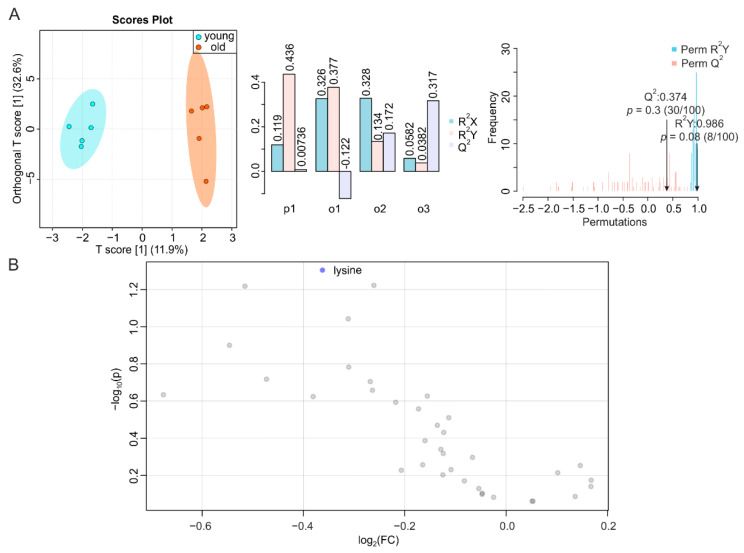
NMR-based metabolomics analysis of duodenum. (**A**) O-PLS-DA plot and cross validation of duodenum samples from young (blue) and old (orange) mice (left), with *n* = 5 each, and permutation between the predictive component (p1) and three orthogonal components (o1, o2, o3) with cross-validated R^2^X, R^2^Y, and Q^2^ coefficients (middle), and histogram showing permutation test with permutation number *n* = 100 (right). (**B**) Volcano plot of differentially regulated metabolites in the duodena of young and aged mice. Decreased (blue) metabolites delimitate the fold change over aging, while grey dots denote insignificantly changed metabolites.

**Figure 6 metabolites-12-00017-f006:**
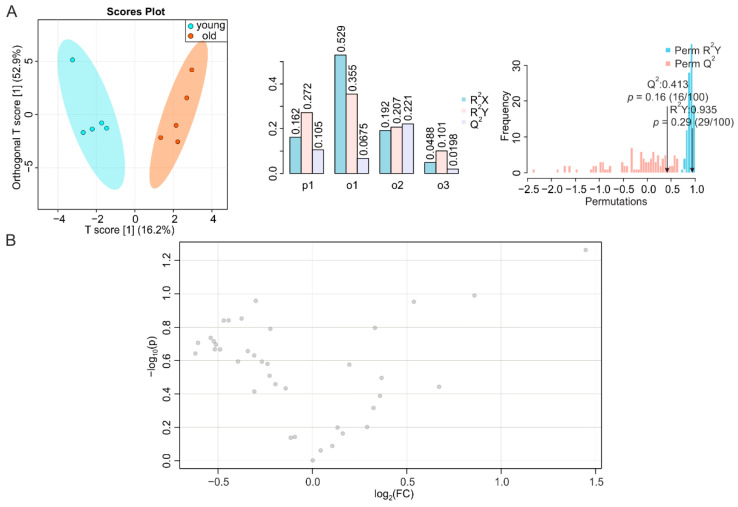
NMR-based metabolomics analysis of jejunum. (**A**) O-PLS-DA plot and cross validation of jejunum samples from young (blue) and old (orange) mice (left), with *n* = 5 each, and permutation between predictive component (p1) and three orthogonal components (o1, o2, o3) with cross-validated R^2^X, R^2^Y, and Q^2^ coefficients (middle), and histogram showing permutation test with permutation number *n* = 100 (right). (**B**) Volcano plot of differentially regulated metabolites in the jejuna of young and aged mice. Grey dots denote insignificantly changed metabolites.

**Figure 7 metabolites-12-00017-f007:**
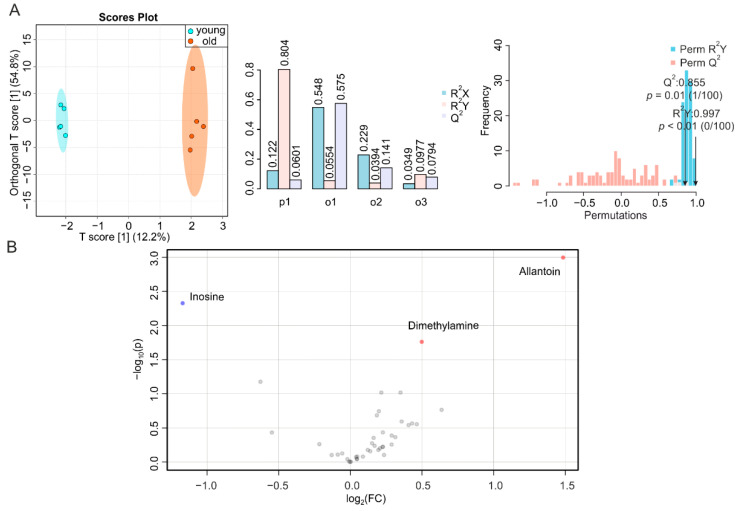
NMR-based metabolomics analysis of ileum. (**A**) O-PLS-DA plot and cross validation of ileum samples from young (blue) and old (orange) mice (left), with *n* = 5 each, and permutation between predictive component (p1) and three orthogonal components (o1, o2, o3) with cross-validated R^2^X, R^2^Y, and Q^2^ coefficients (middle), and histogram showing permutation test with permutation number *n* = 100 (right). (**B**) Volcano plot of differentially regulated metabolites in ilea of young and aged mice. Increased (red) and decreased (blue) metabolites delimitate the fold change over aging, while grey dots denote insignificantly changed metabolites.

**Figure 8 metabolites-12-00017-f008:**
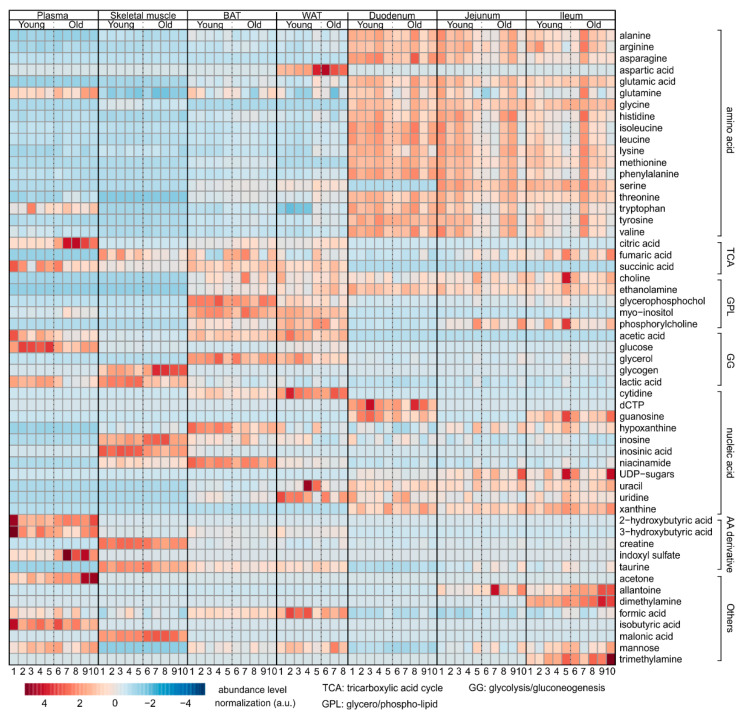
Heat map summarizing the results of NMR metabolomics analysis, indicating the alteration of metabolite levels from the respective tissues. Number of samples is indicated as 1–10 (1–8 for WAT) under each group, while the corresponding metabolites are indicated in each row. Increased metabolites are indicated in red, and decreased metabolites are indicated in blue.

## Data Availability

The data presented in this study are available on request from the corresponding author. The data are not publicly available due to privacy.

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
