# Peer review of "Metabolomic Profiles of Mouse Tissues Reveal an Interplay between Aging and Energy Metabolism"

_metabolites, 2021, doi:10.3390/metabo12010017_

Round 1
Reviewer 1 Report
Metabolomic Profiles of Mice Tissues Reveals an Interplay be-2 tween Aging and Energy Metabolism
This study by Zhou et al investigated aging-related metabolic alterations in tissues responsible for food intake and lipid storage. Authors adopted NMR-based metabolomics approach and profiled plasma, skeletal muscle, brown adipose tissue (BAT), white adipose tissue (WAT), and small intestine parts that include duodenum, jejunum, ileum from young and old mice. These authors found metabolic changes of tissues essential for nutrient uptake and lipid storage. The experimental design is good with the novelty of tissue-specific aging biomarkers. The manuscript draft is written well. Based on the novelty, proper data analysis, and presentation, I would like to recommend this manuscript for publication in metabolites with minor revision.
Introduction
- Line 51: as mass spectrometry and Nuclear Magnetic Resonance (NMR) spectroscopy have been to efficiently characterize the metabolome of an organism
Comment: missing word: as mass spectrometry and Nuclear Magnetic Resonance (NMR) spectroscopy have been used to efficiently characterize the metabolome of an organism.
- Line 126-128: The citing of the figures may confuse the general readers since figure 1A has three figures embedded. It’s not necessary but good to label them individually and cite them in the manuscript,
- Line 127: at 35.1% on the O-PLS-DA plot, indicating that they are well separated
Comment: Please cite the figure
- The increase\decrease or upregulation\downregulation of metabolites have been stated and discussed. The fold change limit for upregulation and downregulation is required in the manuscript.
- Figures 1A and 4A
Comment: Plasma samples from young and old mice, with n=5 each, however, in 1A figure, old mice samples are 4, while in 4A only 3 (can’t feel exactly overlapping point too). Add a statement if an outlier has been removed.
- Figure 8. What software was used to prepare the heatmap? Please include the details.
Author Response
This study by Zhou et al investigated aging-related metabolic alterations in tissues responsible for food intake and lipid storage. Authors adopted NMR-based metabolomics approach and profiled plasma, skeletal muscle, brown adipose tissue (BAT), white adipose tissue (WAT), and small intestine parts that include duodenum, jejunum, ileum from young and old mice. These authors found metabolic changes of tissues essential for nutrient uptake and lipid storage. The experimental design is good with the novelty of tissue-specific aging biomarkers. The manuscript draft is written well. Based on the novelty, proper data analysis, and presentation, I would like to recommend this manuscript for publication in metabolites with minor revision.
We would like to thank the reviewer for her/his very positive feedback.
Line 51: as mass spectrometry and Nuclear Magnetic Resonance (NMR) spectroscopy have been to efficiently characterize the metabolome of an organism
Comment: missing word: as mass spectrometry and Nuclear Magnetic Resonance (NMR) spectroscopy have been used to efficiently characterize the metabolome of an organism.
Thank you, we corrected the error.
Line 126-128: The citing of the figures may confuse the general readers since figure 1A has three figures embedded. It’s not necessary but good to label them individually and cite them in the manuscript,
Line 127: at 35.1% on the O-PLS-DA plot, indicating that they are well separated
Comment: Please cite the figure
We cited the figure and added a more precise link to the subpanels of figure 1A (left, middle and right) to directly link the statements with the observations in figures.
The increase\decrease or upregulation\downregulation of metabolites have been stated and discussed. The fold change limit for upregulation and downregulation is required in the manuscript.
Figures 1A and 4A
We added statements describing the fold changes of the increased/decreased metabolites from each tissue in the corresponding results paragraphs.
Comment: Plasma samples from young and old mice, with n=5 each, however, in 1A figure, old mice samples are 4, while in 4A only 3 (can’t feel exactly overlapping point too). Add a statement if an outlier has been removed.
Two data points of aged samples are overlapped in the O-PLS-DA plot of plasma (orthogonal T score around 0). Two NMR data sets of WAT samples (old mice) could not be processed due technical NMR problems (bad shim). To explain this, we included explanatory statements in the corresponding figure legend and in the materials and methods section, respectively.
Figure 8. What software was used to prepare the heatmap? Please include the details.
The heatmap was generated using Metaboanalyst 5.0. We apologize for the missing information and added it to the materials and methods section.
Reviewer 2 Report
In this work, Zhou et al characterized the tissue-specific metabolic profilings of young and aged mice using NMR, in an effort to identify ageing-related metabolic biomarkers in different tissues and found that intestinal metabolites has no ageing effect, which is really interesting to know.
1. As we know those metabolic markers identification using the line shape fitting and integration of peaks with 1D NMR spectroscopy are challenging because of interfernce of peaks, it is always better for them to validate the results with other techniques like LC-MS/MS. Some minor revisions are recommended.
2. It is really interesting to know that BAT and WAT are different in case of ageing. WAT may relate to obesity that increases with age, did the author found that WAT volume increases in female old mice compared to young ones? I think the metabolomic profile of WAT may be different in case of males? The author may provide little more information on that.
3. The authors found that there is no ageing effect on small intestine based on heat map, but suprisingly, OPLS-DA score plots shows a lot of difference. Can the author explain it?
Author Response
As we know those metabolic markers identification using the line shape fitting and integration of peaks with 1D NMR spectroscopy are challenging because of interfernce of peaks, it is always better for them to validate the results with other techniques like LC-MS/MS. Some minor revisions are recommended.
We thank the reviewer for her/his suggestions and fully agree that metabolomics data of MS and NMR can be cross-validated. To ensure that we do not make any incorrect statements, we decided to integrate only peaks well-separated in the 1H 1D NMR spectra. Using 13C,1H HSQC NMR experiments we further checked that no other metabolite signals overlapped with the signals integrated.
It is really interesting to know that BAT and WAT are different in case of ageing. WAT may relate to obesity that increases with age, did the author found that WAT volume increases in female old mice compared to young ones? I think the metabolomic profile of WAT may be different in case of males? The author may provide little more information on that.
Thank you for pointing this out to us. Indeed, it will be interesting to address this in a follow-up study.
Although not determined in this study, it has been shown that aging‐induced body mass gain in female mice is associated with increased WAT but not BAT mass (doi: 10.1111/iep.12228). The authors also showed that female mice exhibit a progressive age‐dependent loss of subcutaneous and visceral WAT browning, whereas BAT changes toward a fat storage phenotype.
We agree with the reviewer that the metabolomic profile of WAT may be different in males. A recent study suggested that increased UCP1 expression in WAT of female mice is a key factor that underlies sex differences in metabolic phenotypes by affecting mitochondrial function (doi: 10.1016/j.mce.2021.111337).
We have added the following statement to the revised version of the manuscript:
“Although not determined in this study, it has been shown that aging‐induced body mass gain in female mice is associated with increased WAT but not BAT mass (cite: doi: 10.1111/iep.12228). The authors also showed that female mice exhibit a progressive age‐dependent loss of subcutaneous and visceral WAT browning, whereas BAT changes toward a fat storage phenotype. Despite these differences in WAT browning, function and depot capacity during aging in humans and mice (REF 24, 38 doi: 10.1111/iep.12228) , the metabolome of WAT was comparable between young and old mice.”
The authors found that there is no ageing effect on small intestine based on heat map, but suprisingly, OPLS-DA score plots shows a lot of difference. Can the author explain it?
In the O-PLS-DA plot, the predictive component score (T score) is low for the small intestine samples, and this indicates that only few variations allow to separate the 2 groups. In order to validate the O-PLS-DA model, we carry out cross-validation and the low Q2 values and p values higher than 0.05, especially for duodenum and jejunum, showed that the model is insufficiently predictive. In line with this, the univariate analyses (volcano plots) showed no or barely significantly changed metabolites.